# Global turnover of soil mineral-associated and particulate organic carbon

Zhenghu Zhou [1,2], Chengjie Ren [3] ✉, Chuankuan Wang [1], Manuel Delgado-Baquerizo [4], Yiqi Luo[5], Zhongkui Luo [6], Zhenggang Du[1], Biao Zhu [7], Yuanhe Yang [8], Shuo Jiao [9], Fazhu Zhao [10], Andong Cai [11], Gaihe Yang[3] & Gehong Wei [9] ✉

Soil organic carbon (SOC) persistence is predominantly governed by mineral protection, consequently, soil mineral-associated (MAOC) and particulate organic carbon (POC) turnovers have different impacts on the vulnerability of SOC to climate change. Here, we generate the global MAOC and POC maps using 8341 observations and then infer the turnover times of MAOC and POC by a data-model integration approach. Global MAOC and POC storages are $975^{987}_{964}$ Pg C (mean with 5% and 95% quantiles) and $330^{337}_{323}$ Pg C, while global mean MAOC and POC turnover times are $129^{383}_{45}$ yr and $23^{82}_{5}$ yr in the top meter, respectively. Climate warming-induced acceleration of MAOC and POC decomposition is greater in subsoil than that in topsoil. Overall, the global atlas of MAOC and POC turnover, together with the global distributions of MAOC and POC stocks, provide a benchmark for Earth system models to diagnose SOC-climate change feedback.

Soil organic carbon (SOC) is the property of terrestrial ecosystems due to its multi-functionality[1]. Positive feedback to climate change would occur if SOC is transferred to the atmosphere by a warming-induced acceleration of its decomposition[2]. Recent advances attest that the persistence of SOC to microbial decomposition is predominantly controlled by physical accessibility and isolation rather than its biochemical recalcitrance[3–6]. Separating SOC into mineral-associated (MAOC) and particulate organic carbon (POC) consequently enables a more accurate prediction of soil vulnerability to climate change[7,8]. Most Earth system models with unmeasurable Century and RothC carbon pools place low confidence on both spatial-temporal SOC

distribution and projection to climate change[9–11]. Incorporating the measurable and biophysically defined MAOC and POC pools into Earth system models probably is an optimal approach to trace the fate of SOC under global change[12–15] and would be the next generation of soil carbon cycling model[16,17].

The critical drivers and global distributions of MAOC and POC are not well quantified, and the corresponding data product with high accuracy is also lacking for modeling community. For example, Georgiou et al.[18] have created a global map of MAOC but excluding much of the boreal region with a small dataset (1144 soil observations). Synthesizing 162 soil observations, a recent study suggested that topsoil in

[1]College of Ecology and Key Laboratory of Sustainable Forest Ecosystem Management-Ministry of Education, Northeast Forestry University, Harbin, Heilongjiang, China. [2]Northeast Asia Biodiversity Research Center, Northeast Forestry University, Harbin, Heilongjiang, China. [3]State Key Laboratory for Crop Stress Resistance and High-Efficiency Production, College of Agronomy, Northwest A&F University, Yangling 712100 Shaanxi, China. [4]Laboratorio de Biodiversidad y Funcionamiento Ecosistémico. Instituto de Recursos Naturales y Agrobiología de Sevilla (IRNAS), CSIC, Av. Reina Mercedes, Sevilla, Spain. [5]School of Integrative Plant Science, Cornell University, Ithaca, NY, USA. [6]College of Environmental and Resource Sciences, Zhejiang University, Hangzhou, China. [7]Institute of Ecology, College of Urban and Environmental Sciences, and Key Laboratory for Earth Surface Processes of the Ministry of Education, Peking University, Beijing, China. [8]State Key Laboratory of Vegetation and Environmental Change, Institute of Botany, Chinese Academy of Sciences, Beijing, China. [9]State Key Laboratory for Crop Stress Resistance and High-Efficiency Production, College of Life Sciences, Northwest A&F University, Yangling 712100 Shaanxi, China. [10]Shaanxi Key Laboratory of Earth Surface System and Environmental Carrying Capacity, Northwest University, Xi'an, Shaanxi, China. [11]Institute of Environment and Sustainable Development in Agriculture, Chinese Academy of Agricultural Sciences, Beijing, China. ✉e-mail: Rencj1991@nwsuaf.edu.cn; weigehong@nwsuaf.edu.cn

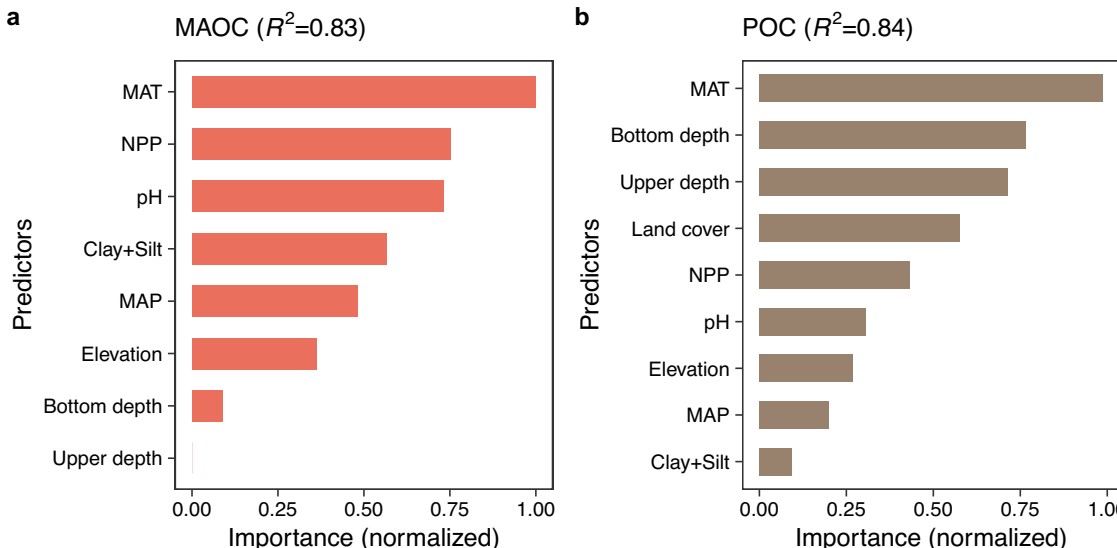

**Fig. 1 | The importance of predictors for mineral-associated and particulate organic carbon. a** The importance of predictors for mineral-associated organic carbon (MAOC). **b** The importance of predictors for particulate organic carbon (POC). Two types of importance from random forest algorithms (impurity and permutation) were first normalized to the interval of [0, 1] according to their maximum and minimum values. The mean of the normalized importance was then calculated. MAT mean annual temperature, MAP mean annual precipitation, NPP net primary productivity. Source data are provided as a Source Data file.

cold regions was distributed predominantly by more vulnerable POC rather than by MAOC[8]. MAOC with simple molecular compounds persists in soils over hundreds of years and is predominantly protected by minerals via van der Waals force, hydrogen bond, covalent bond, complexation, cation bridging, and pore sorption[19,20]; while POC with complex molecular compounds persists in soils up to decades relying on biochemical recalcitrance and aggregate protection[7,19]. Therefore, the global distributions of MAOC and POC may be dependent on different factors. SOC storage capacity can be given as the product of carbon influx and turnover time[21]. Assessing SOC turnover time can offer an insight into the SOC dynamics under environmental changes[22], which is estimated via multiple approaches, such as radiocarbon[23], remote sensing[24,25], and Earth-system models[26]. In addition, subsoils (>20 cm) contain more than half of global SOC stocks[27]. However, the uncertainty of the response of SOC to climate warming increases with soil depth because the mechanisms that control deeper SOC's turnover are still one of the largest challenges for soil carbon cycling[28–30]. Although MAOC and POC turnovers have very different impacts on soil fertility and carbon emission[7], the global distributions of MAOC and POC turnover times across spatial and soil profiles, and their covariations with climate, topography, vegetation, and edaphic factors have not been quantified.

Here, we synthesized 8341 soil observations that separate the bulk SOC into MAOC and POC fractions globally (Supplementary Fig. 1 and 2). The current datasets spanned broadly along spatial and climatic gradients: latitude ranged from −63° S to 69° N, longitude ranged from −155° S to 153° N, mean annual temperature (MAT) ranged from −9 °C to 29 °C, while mean annual precipitation ranged from 58 mm to 3128 mm. We aimed to produce global maps of MAOC and POC by a machine learning approach, and to reveal the global spatial and profile distributions of turnover times of MAOC and POC using a data-assimilation approach.

We generate the MAOC and POC maps by soil layers of 0–20, 20–40, 40–60, 60–80, and 80–100 cm using random forest model with high accuracy ($R^2 > 0.83$). Global MAOC and POC storages are $975^{987}_{964}$ Pg C (mean with 5% and 95% quantiles from 100-times bootstrapping) and $330^{337}_{323}$ Pg C in the top meter, respectively. We also propose a two-pool model including the measurable MAOC and POC to infer their turnover times along the soil profile. The global mean MAOC and POC turnover times in the top meter are $129^{383}_{45}$ yr and $23^{82}_{5}$ yr, respectively. MAT is the most important factor for MAOC and POC turnovers and their distribution along the soil profile. Climate warming-induced acceleration of MAOC and POC decomposition is greater in subsoil than that in topsoil. Our global estimations of the stocks and turnover times of MAOC and POC would provide a benchmark for Earth system models to diagnose SOC-climate change feedback.

## Results and Discussion

### Global distributions of mineral-associated and particulate organic carbon

Geographical distributions of MAOC and POC were driven by multiple climate, topography, vegetation, and edaphic factors (Figs. 1 and 2; Supplementary Fig. 3). MAT, mean annual precipitation, net primary productivity, elevation, pH, clay plus silt content, and sampling depth were the best predictors of MAOC, which explained 83% variance (Fig. 1). Land cover was also important for the spatial variation of POC, together with the above factors for MAOC, which explained 84% variance for POC (Fig. 1). As expected, soils with greater carbon inputs (higher net primary productivity) and lower decomposition rate (lower MAT) have greater MAOC and POC (Fig. 2). There is controversy over whether MAOC is saturated[31,32]. The objectors of the concept of MAOC saturation suggested that organic matter can bind to other organic matter bonded to minerals, in unlimited "skyscraper-like" structures[31]. Here, the partial dependence plot showed that there was a saturation relationship between MAOC and net primary productivity (carbon inputs) considering the multiple covariates (Fig. 2). Therefore, our global dataset supported the concept of mineral-associated organic matter saturation.

Both MAOC and POC were higher in acidic soils (Fig. 2) because microbial activities are generally inhibited by low pH[33,34]. In addition, soil pH was the second important predictor for the spatial variation of MAOC, because soil pH mediates many geochemical processes[35]. Especially, carbon sorption onto soil minerals is pH-dependent, and mineral surfaces become less positive and organic ligands become more negative at higher pH, which would minimize the carbon sorption to mineral surfaces[14,20,36–38]. The mechanisms of SOC stabilization shift with rising pH from predominantly organo-metal complexation,

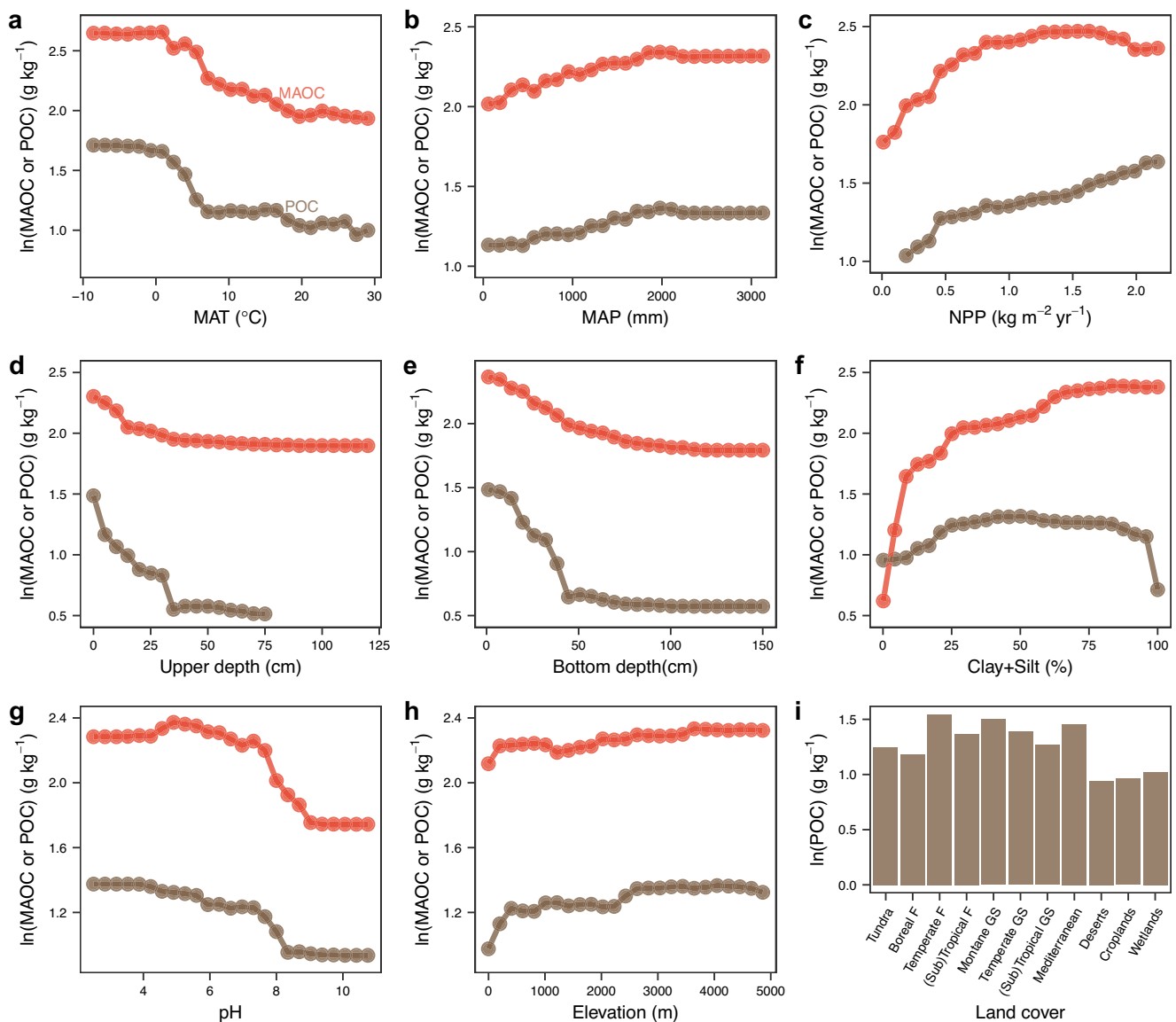

**Fig. 2 | Partial dependences showing the effects of predictors on mineral-associated and particulate organic carbon. a** Partial dependences for mean annual temperature (MAT). **b** Partial dependences for mean annual precipitation (MAP). **c** Partial dependences for net primary productivity (NPP). **d** Partial dependences for upper sampling depth. **e** Partial dependences for bottom sampling depth. **f** Partial dependences for clay plus silt content (clay+silt). **g** Partial dependences for pH. **h** Partial dependences for elevation. **i** Partial dependences for land cover. F forests, GS grasslands and shrublands. Source data are provided as a Source Data file.

to association with short-range-order phases, to Ca complexation and cation bridging with phyllosilicates, increasing soil pH consequently decreases the accumulation of MAOC[38]. Soil clay- and silt-size minerals provide the scaffolds and habitats for MAOC, together with the inhabitation of high clay plus silt content on microbial decomposition by other mechanisms, soil clay plus silt had positive effect on MAOC (Fig. 2). In addition, soil minerals can protect POC from microbial decomposition[39], and a recent study had provided the visible evidence that occlusion of organic matter into aggregates and formation of organo-mineral associations occur concurrently on POC[40]. Land cover was important for POC but not for MAOC (Fig. 2 and Supplementary Fig. 3), implying that the quality of litter may have a weak influence on the formation of MAOC at a global scale. In addition, POC may be more sensitive to the conversion of natural land cover types to croplands than MAOC.

Finally, the 10-fold validation showed that the random forest had good performance in predicting the global distributions of MAOC and

POC, with $R^2$ of 0.92 and 0.93, respectively (Supplementary Fig. 4). We generated the MAOC and POC maps by soil layers of 0–20, 20–40, 40–60, 60–80, and 80–100 cm, respectively (Fig. 3; Supplementary Figs. 5 and 6). POC showed a shallower distribution than MAOC because POC was more dependent upon carbon inputs than soil properties (Figs. 1 and 3). We derived a global MAOC storage of $975^{987}_{964}$ Pg C and POC storage of $330^{337}_{323}$ Pg C in the top meter (Fig. 3). Totally, the global SOC storage is $1306^{1321}_{1292}$ Pg C, which is well consistent with previous estimations from observations (ranged from 504 Pg C to 1849 Pg C)[41] and modeled methods (ranged from 510 Pg C to 3040 Pg C)[11].

## Global turnover of mineral-associated and particulate organic carbon

Using a proposed two-pool model and a data assimilation approach (see methods), the global mean MAOC and POC turnover times in the top meter of soils were estimated as $129^{383}_{45}$ yr and $23^{82}_5$ yr, respectively (Fig. 4). MAT is the most important factor for MAOC and POC

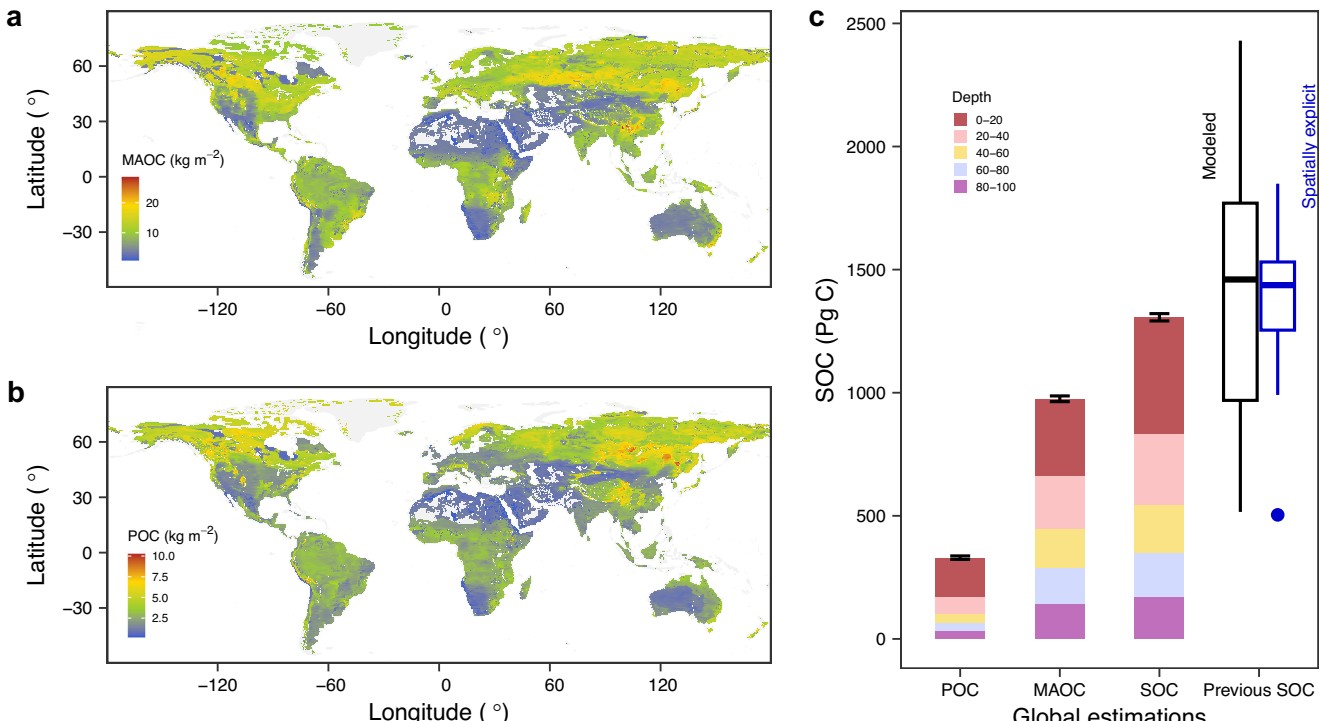

**Fig. 3 | Global estimations of mineral-associated and particulate organic carbon storages. a** Global map of mineral-associated organic carbon (MAOC) storage (0–100 cm). **b** Global map of particulate organic carbon (POC) storage (0–100 cm). **c** Global soil organic carbon (SOC) storages from our estimation (bars) and previous estimations from observations[41] (blue boxplot) and modeled methods[11] (black boxplot). The error bars are 5% and 95% quantiles from 100-times bootstrapping. Center line, median. Box limits, upper and lower quartiles. Whiskers, 1.5 times interquartile range. Source data are provided as a Source Data file.

turnovers (Supplementary Figs. 7 and 8), supporting a previous viewpoint that temperature dominates the SOC turnover[22,26]. Semi-arid, dry sub-humid, and humid regions had similar turnover-MAT relationships, but differed from the relationship in arid regions (Supplementary Fig. 8). Therefore, moisture can dominate the turnover in places that are highly water-limited, such as the long-term turnover times of MAOC and POC in desert regions (Fig. 4).

The turnover times of MAOC and POC were increased as increased soil depth (Fig. 4). Referring to the Community Land Model (CLM)[42], the reduced rate of decomposition rate with soil depth in our two-pool model was regulated by an e-folding depth for carbon turnover. A greater value of e-folding depth for carbon turnover represents a slower decrease in decomposition rates with increased soil depth, while a smaller value of e-folding depth for carbon turnover represents a more rapid one (see methods). MAT predominately controls the variations of e-folding depths for MAOC and POC turnovers (Fig. 5). First, cold land cover (tundra and boral forest) had shallower root distribution than warm ones (such as (sub)tropical forests and grasslands)[43]. The absence of fresh carbon is suggested to be a critical mechanism that prevents the decomposition in deep soil layers because microbial activity was limited by an essential source of energy[44,45]. Second, the slow SOC decomposition at subsoil could result from a lack of oxygen. Physically, warm climate would enhance the gaseous diffusivity between subsoil and atmosphere[46], subsoils in (sub)tropical regions consequently had lower oxygen limitation than that in cold regions. We also found that the e-folding depth for carbon turnover for POC was about 1.7 times as large as that of MAOC, indicating that an increase in soil depth had stronger inhibition on MAOC decomposition than that on POC. The total mineral surface is one of the most limiting factors influencing SOC bound to soil minerals. Compared with topsoil, subsoils had a greater carbon saturation deficit, while a greater carbon saturation deficit not only increases the stabilization of MAOC but also increases the transfer efficiency from POC to MAOC[47].

## Limitations and uncertainties

There are several limitations and uncertainties in the current study. First, like the limitation for all global synthesis, the soils were sampled by different methods, time, successional stage, repeatability, and so on, which would result in uncertainties. Second, the assumption of steady was suggested to improve the operability of data assimilation[48,49]. Although SOC in ecosystems with anthropogenic disturbance, such as degraded and restored ecosystems, may not be at a steady state, previous studies have shown that such a disequilibrium component is minor for SOC pools considering its long turnover time[50,51]. However, such an assumption of a steady state indeed introduces uncertainties. Third, the data points in Russia, Canada, and Africa are relatively small, which may limit the accuracy of our estimates of MAOC and POC in these regions, especially for permafrost in Russia and Canada, which are rich in SOC.

Despite the above limitations, to our knowledge, we presented the first global atlas of MAOC and POC turnover times using a big dataset of 8341 soil observations, together with the global distributions of MAOC and POC stocks, which provided a benchmark for Earth system models. Our findings supported that temperature dominantly controls both storage and turnover times of MAOC and POC. In addition, the positive effects of MAT on e-folding depths for both MAOC and POC turnovers imply that climate warming-induced increase in decomposition was greater in subsoil than that in topsoil. Therefore, the loss of MAOC and POC in deep soils may have positive feedback to climate change, it is urgent to prevent the SOC loss from deep soils.

## Methods
### Data collection
An extensive literature survey was conducted through the Google Scholar and China National Knowledge Infrastructure databases until 2024 using keywords of "soil" "mineral-associated" and "carbon".

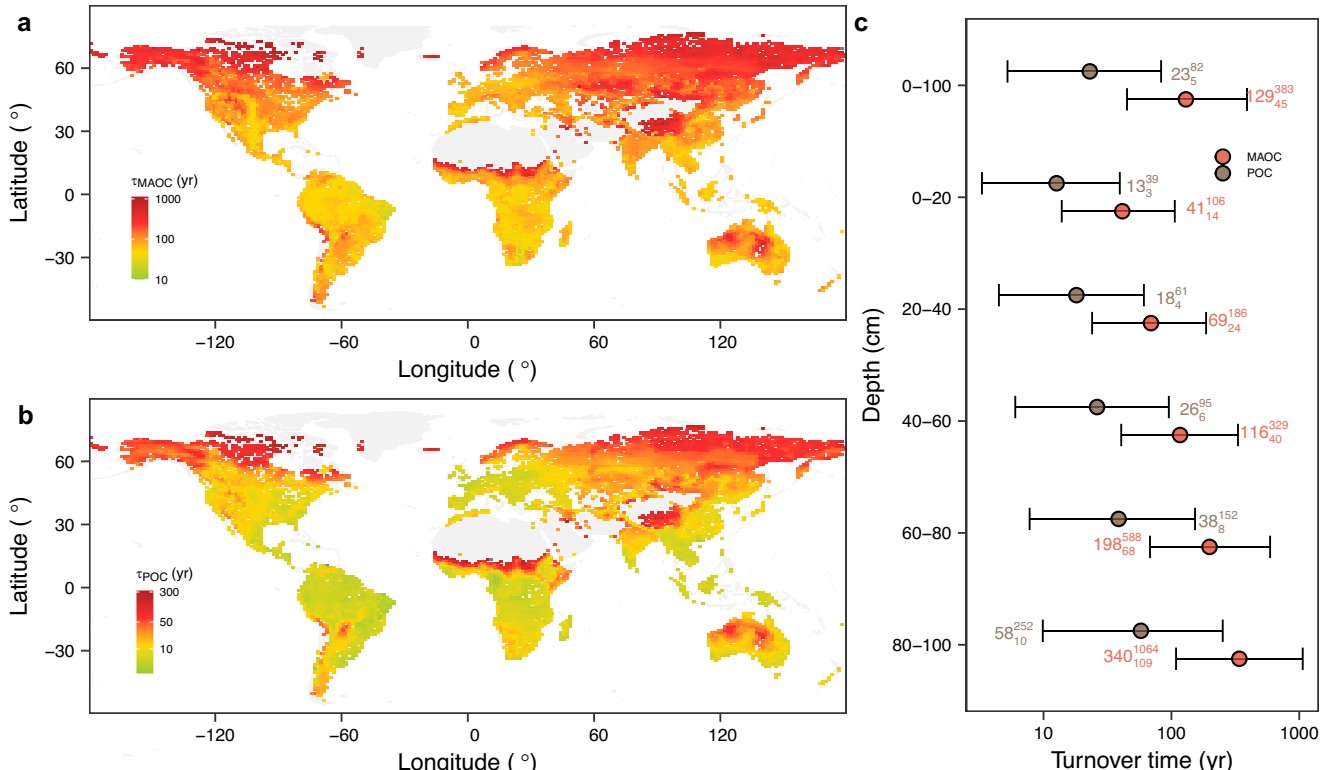

**Fig. 4 | Spatial and profile patterns of turnover times of mineral-associated and particulate organic carbon. a** mineral-associated organic carbon (MAOC) turnover time (0–100 cm). **b** particulate organic carbon (POC) turnover time (0–100 cm). **c** Profile patterns of turnover times of MAOC and POC. The error bars are 5% and 95% quantiles from 100-times bootstrapping. Source data are provided as a Source Data file.

Differing from a recent global synthesis that collected MAOC passed through 63-μm sieve (1144 entries)[18], but the sieve size varied greatly across all the entries, where 53-μm sieve size has the largest proportion (60.1%), followed by 20-μm sieve size (24.1%) and 63-μm sieve size (7.8%). To reduce the sieve size impacts from different entries, our datasets just recorded the MAOC passed through 53-μm sieve. We only included studies of synchronous reporting SOC, POC, and MAOC. Finally, our datasets included 8341 entries spanning from 1990 to 2022 (Supplementary Figs. 1 and 2).

To quantify the drivers of global distributions of POC and MAOC, we consider the effects of climate (MAT and mean annual precipitation), topography (elevation and slope), vegetation (net primary productivity and land cover types), soil properties (soil types, pH, clay plus silt content, base saturation, and cation exchange capacity), nitrogen deposition, sampling depth and time. These covariates were either recorded from the original studies or obtained from global datasets (Supplementary Table 1). The land cover was classified into tundra, boreal forest, temperate conifer forest, temperate broadleaf/mixed forest, (sub)tropical moist forest, (sub)tropical dry forest, montane grassland/shrubland, temperate grassland/shrubland, (sub)tropical grassland/shrubland, Mediterranean, desert, wetland, and cropland according to the Terrestrial Ecoregions of the World and the MODIS Land Cover product. Most of the data entries were from 2000–2021 (Supplementary Fig. 2), together with the time range of MODIS net primary productivity, we calculated the average annual temperature, precipitation, and net primary productivity from 2001 to 2021. Therefore, we kept the consistent times for climates, net primary productivity, MAOC, and POC.

**Mapping global mineral-associated and particulate organic carbon.** First, we compared the performances of six typical machine learning algorithms, including random forest, extreme gradient boosting, support vector machines, recursive partitioning and regression trees, neural networks, and multivariable linear regression. We found that the random forest had the least root mean square error (Supplementary Table 2), therefore, random forest was used to general the global maps of MAOC and POC.

Second, semivariance was first calculated to explore the degree of spatial auto-correlation of MAOC and POC contents, while the semi-variograms did not show strong spatial auto-correlation (Supplementary Fig. 9). In addition, Moran's index was also used to test the significance of spatial autocorrelation in random forest models. A greater significantly positive Moran's index indicates that there is spatial autocorrelation for a given variable at a specific distance threshold. Here, we did not find a significant spatial autocorrelation for MAOC and POC concentrations in random forest models (Supplementary Fig. 9).

Third, the "wrapper" method[52] was used to select the covariates among climate (MAT and MAP), topography (elevation and slope), vegetation (net primary productivity and land cover types), soil properties (soil types, pH, clay plus silt content, base saturation, and cation exchange capacity), nitrogen deposition, sampling depth and time. Specifically, by re-calibrating a random forest model several times, each time removing the least important covariate, one may expect to considerably reduce the overall number of covariates with little or no decrease in model prediction accuracy. We used methods of impurity and permutation in the ranger function of the "ranger" package to quantify the importance of predictors, we then normalized the two types of importance to the interval of [0, 1] according to their maximum and minimum values. The mean of the normalized importance was used. The final random forest model for MAOC included MAT, mean annual precipitation, net primary productivity,

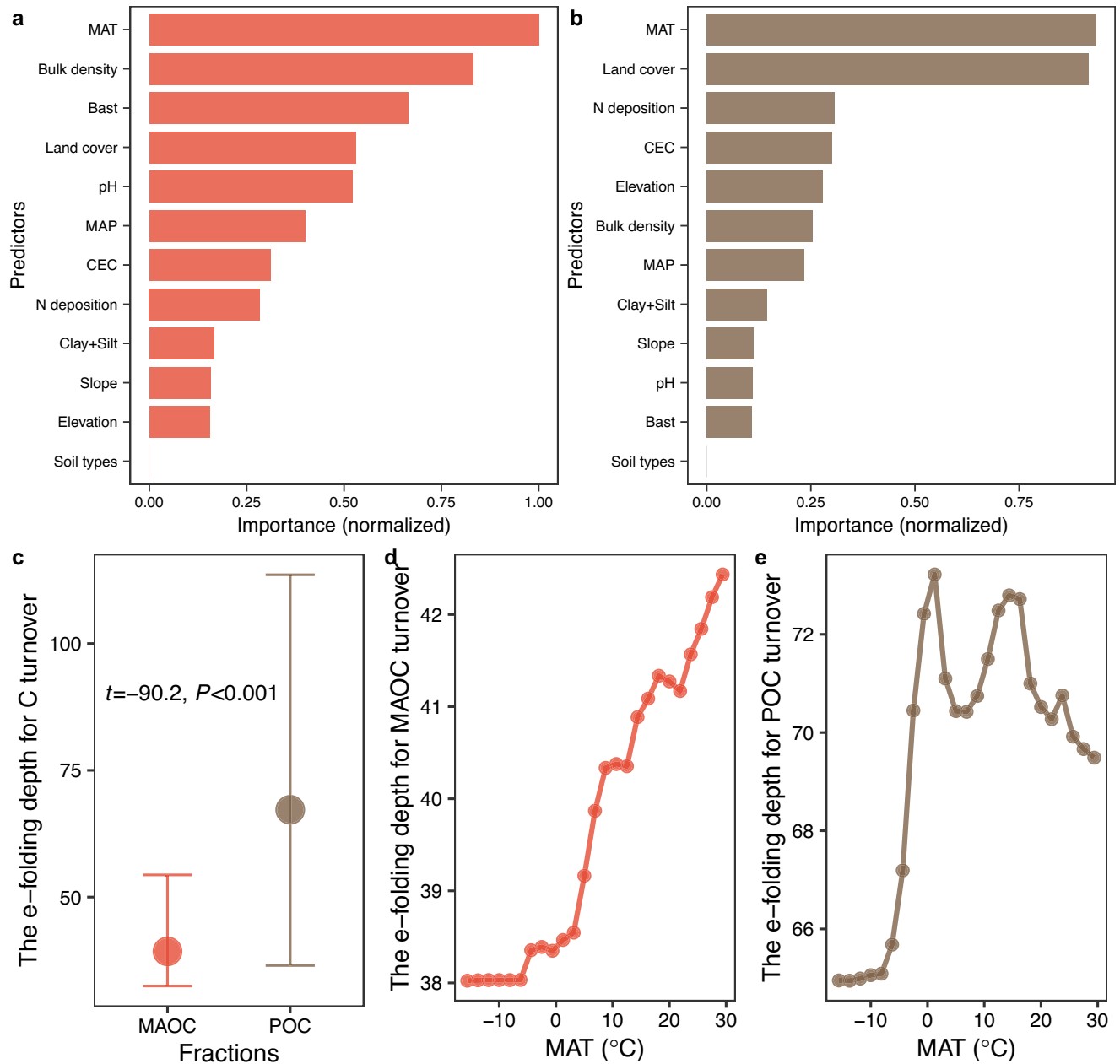

**Fig. 5 | Factors influencing the reduced rate of decomposition rate with soil depth. a** and **b** Importance of predictors for e-folding depths for mineral-associated (MAOC) and particulate organic carbon (POC) turnovers, respectively. A greater value of e-folding depth for carbon turnover represents a slower reduced rate of decomposition rate with increased soil depth, while a smaller value of e-folding depth for carbon turnover represents a more rapid one. Two types of importance (impurity and permutation) were first normalized to the interval of [0, 1] according to their maximum and minimum values. The mean of the normalized importance was then calculated. **c** The e-folding depths for MAOC vs. POC turnovers (mean with 5% and 95% quantiles) with paired *t*-test. **d** and **e** Partial dependence showing the effect of mean annual temperature (MAT) on e-folding depths for MAOC and POC turnovers, respectively. MAP mean annual precipitation. NPP net primary productivity. CEC cation exchange capacity. Bast base saturation. Source data are provided as a Source Data file.

elevation, pH, clay plus silt content, and sampling depth, while the final random forest model for POC included MAT, mean annual precipitation, net primary productivity, land cover, elevation, pH, clay plus silt content, and sampling depth (Fig. 1 and Supplementary Fig. 3).

Furthermore, *K*-fold cross-validation (with *K* = 10) was used to validate the random forest models, and our random forest model showed good performance (Supplementary Fig. 4). To generate the spatially-explicit and quantitative maps of MAOC and POC contents, the global predictors were aggregated or resampled at a resolution of 0.5° × 0.5° by soil layers (0–20, 20–40, 40–60, 60–80, and 80–100 cm). The 100-time bootstrapping was used to generate the uncertainties of global maps (for each bootstrapping, 90% of the data were randomly sampled).

**Model desperation.** A two-pool model of MAOC and POC was proposed following the framework Century (Supplementary Fig. 10):

$$\frac{dPOC_i(t)}{dt} = I_i + MAOC_i KM_i T_{MAOCtoPOC} - POC_i KP_i + POC_{i+1}D - POC_i A$$

(1)

$$\frac{\mathrm{d}MAOC_i(t)}{\mathrm{d}t} = POC_i KP_i T_{POCtoMAOC} - MAOC_i KM_i \\ + MAOC_{i+1} D - MAOC_i A \qquad (2)$$

where $i$ is $i$th soil depth, referring to 0–20, 20–40, 40–60, 60–80, and 80–100 cm, respectively. $I_i$ is net primary productivity allocation into $i$th soil depth, which refers to a recent global study that has quantified the allocation of net primary productivity into belowground by the same soil depth[43]. Therefore, we firstly obtained the averaged MODIS net primary productivity from 2001 to 2023. We then calculated the net primary productivity allocation into 0–20, 20–40, 40–60, 60–80, and 80–100 cm, respectively. $MAOC_i$ and $POC_i$ are MAOC and POC pools of $i$th soil depth. $KM_i$ and $KP_i$ are the decomposition rates (inverse of turnover time) of MAOC and POC, respectively. $T_{MAOCtoPOC}$ is SOC transformation from MAOC to POC, while $T_{POCtoMAOC}$ is SOC transformation from POC to MAOC. $D$ and $A$ represent the diffusion and advection, respectively.

Referring to Community Land Model (CLM)[42], an exponential decrease in decomposition was used to account for the regulation of soil depth on decomposition:

$$KM_i = KM_1 \exp\left(-\frac{z_i - 10}{z_M}\right) \qquad (3)$$

$$KP_i = KP_1 \exp\left(-\frac{z_i - 10}{z_P}\right) \qquad (4)$$

where $z_M$ and $z_P$ are the e-folding depths for carbon turnovers for MAOC and POC, respectively. $z_i$ is the soil depth of $i$th layer.

**Data assimilation.** We have more robust global estimates of productivity than of heterotrophic respiration. Therefore, if we assume that soils are approximately at a steady state, we can analytically calculate the steady-state pool sizes of MAOC and POC by letting Eqs. (1, 2) equal zero, i.e., a semi-analytical solution to accelerate spin-up[53]. Bayesian probabilistic inversion was used to calibrate the parameters in the two-pools model[9,54]. In Bayes' theorem, the posterior probability density functions ($P(\theta|Z)$) of model parameters ($\theta$) can be obtained from prior probability density functions of the parameters ($P(\theta)$) and the likelihood function ($P(Z|\theta)$):

$$P(\theta|Z) \propto P(Z|\theta)P(\theta) \qquad (5)$$

The $P(Z|\theta)$ was calculated with the assumption that errors between observed and modeled C pools were independent of each other and followed a multivariate Gaussian distribution with a zero mean:

$$P(Z|\theta) \propto \exp\left\{\frac{(C_{obs} - C_{mod})^2}{2\sigma^2}\right\} \qquad (6)$$

where $C_{obs}$ and $C_{mod}$ are the observed and modeled carbon pools of soil layers of 0–20, 20–40, 40–60, 60–80, and 80–100 cm, while $\sigma$ is the standard deviation of observation.

To generate the posterior distributions of model parameters, we first specified the prior ranges of the model parameters in Supplementary Table 3 based on the literature, educated guess, and hypothesis testing[54]. Specifically, a previous opinion study proposed that POC and MAOC have vague turnover times of <10 years to decades and decades to centuries, respectively. Here, the wide ranges of turnover times were set (Supplementary Table 3). The e-folding depths were set as 50 cm in CLM referring to one study of $^{14}$C profiles[42], therefore, we set a range of 30–200 cm. The current vertically resolved models, such as CLM[42] and ORCHIDEE-SOM[55], simulate the vertical transformation of

SOC using two processes of diffusion and advection. However, these two fluxes were not well quantified in fields with diverse conditions. We proposed a relatively wide of range for these two processes by referring to CLM[42] and ORCHIDEE-SOM[55]. Finally, according to the global distributions of diffusion and advection optimized by the following processes, our original set was acceptable for diffusion and advection (not clustered to the upper limit; Supplementary Fig. 11).

Once we specified the parameter ranges, the probabilistic inversion was carried on using the Metropolise-Hastings algorithm, a Markov Chain Monte Carlo technique, to construct the $P(\theta|Z)$[56]. Briefly, the Metropolise-Hastings algorithm includes a proposing step and a moving step. In the proposing step, a new parameter set $\theta^{new}$ is proposed based on the previously accepted parameter set $\theta^{old}$ and a proposal distribution:

$$\theta^{new} = \theta^{old} + \frac{r \times (\theta_{max} - \theta_{min})}{D} \qquad (7)$$

where $\theta_{max}$ and $\theta_{min}$ are the maximum and minimum values of parameters, $r$ is a random variable between -0.5 and 0.5, and D is used to control the proposed step size and was set to 5[56]. In the moving step, $\theta^{new}$ was tested against the Metropolis criterion to examine if the new parameter set should be accepted or rejected. The first half of the accepted samples were discarded and only the rest were used to generate $P(\theta|Z)$, i.e., these optimized parameters had the greatest accuracy in predicting the MAOC and POC dynamics (Supplementary Fig. 10).

The turnover times of MAOC ($\tau_{MAOC_i}$) and POC ($\tau_{POC_i}$) in $i$th layers are the reciprocal of corresponding decay constants:

$$\tau_{MAOC_i} = \frac{1}{KM_i} \qquad (8)$$

$$\tau_{POC_i} = \frac{1}{KP_i} \qquad (9)$$

The two-pool model was conducted with a spatial resolution of 1.5° × 1°, i.e., we optimized model parameters for each grid by observing profile distributions of MAOC and POC. The turnover time of the whole profile (0–100 cm) can be calculated as the weighted mean of turnover time by pool sizes of different soil layers. We also used random forests to explore the regulation of climate, topography, vegetation, soil properties, and nitrogen deposition on turnover times and the e-folding depth of intrinsic decomposition rates.

All the statistical analyses were performed in R 4.2.2 (R Core Team 2022).

## Data availability
The code that supports the findings of this study are openly available in figshare at https://figshare.com/account/articles/25962769. Source data are provided with this paper.

## Code availability
The code that supports the findings of this study are openly available in figshare at https://figshare.com/account/articles/25962721.

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

## Acknowledgements
We thank Zheng Shi and Yuanyuan Huang for their suggestions on this study. We thank all the researchers whose data were used in this global synthesis. This work was financially supported by the National Key Research and Development Program of China (2021YFD2200401), the National Natural Science Foundation of China (42377345), and the Fundamental Research Funds for the Central Universities (2572021CG07).

## Author contributions
C.R., Z.Z. and G.W. conceived and designed the project; C.R., S. J., A. C., Y. Y. and F. Z. collected the data; Z.Z., B. Z., M. D. and C.R. conducted the data analyses with help from Y.L., Z. D., G.Y., C.W. and Z.K.; Z.Z. pro-posed the new two-pool model; Z.Z. and C.R. wrote the manuscript; All authors contributed to discussing the results, writing, and editing the paper.

## Competing interests
The authors declare no competing interests.
