## [Peer Review File · Nature Communications]

Global turnover of soil mineral-associated and particulate organic carbonEditorial Note: Parts of this Peer Review File have been redacted as indicated to remove third-party material where no permission to publish could be obtained.

REVIEWER COMMENTS

Reviewer #1 (Remarks to the Author):

The authors conducted an impressive study which they applied a machine learning algorithm with 7163 soil observations combined with a data-model to generate the global MAOC and POC maps, as well as the turnover times of MAOC and POC. The results suggest that turnover times of MAOC and POC are negatively affected by the mean annual temperature, with POC turnover more sensitive to temperature in warm regions but less in cold regions compared to the MAOC turnover. These results are applaudable and will constitute a valuable database for the research community. However, there are some major concerns (particularly in Method section) that need clarification.

Detailed comments

L29, “...the vulnerability of soil carbon to warming.”, not only to warming, change as “...the vulnerability of soil carbon to climate change.

L52-58, the fates of MAOC and POC are indeed important for Earth system models, but it was not a key point for this paper. I suggest one sentence is enough for expressing the implication of the MAOC and POC to Earth system models

L40-71: the two paragraphs are not in good logic; authors did not focus on the knowledge gap in relation with the results later presented. For instance, how climate, plant, soil pH and microbe affect the MAOC and POC pool and turn over? Authors did not mention in this section, but authors talked much more about their effects on the MAOC and POC dynamics in later parts.

L159, please provide a detailed proportion for “most of studies”. Also, I think the authors should provide a scientific mechanism about selecting 53- μm sieve rather than 63- μm sieve.

How do you detail with temporal variation of soil sampling? Keep consistent time is important for mapping spatial trends of MAOC and POC. The same issue for the predictors.

L182, there are many machine learning algorithms, why do you select the random forest? How do you identify the optimal hyper-parameters? Please provide the details. How do you evaluate the uncertainties of global maps?

L199-229, What are the purpose of this section, and the linkages with previous sections in the methods? Why not use the in-situ data in two-pool model? How do you identify the parameters (e.g., TE) of two-pool model? $R^2 = 0.987$ is very high, how do you avoid the overfitting?

Are results consistent for the in-situ data and incubation data? Caution must be exercised for the translation of incubation analyses, which are the conditions used so far for most research, to realistic field conditions.

L231-232, I note that the authors did not consider the anthropogenic factors. If consider the anthropogenic factors, here the assumption is wrong. However, the author did not clarify this, for both

global maps and two-pool model.

L255, how do you identify the reference decomposition rate and reference temperature? Please provide the details. How do you perform the translation of incubation analyses to field conditions?

L271, please provide the references to support your assumption (linear model). Because Q10 is traditionally represented by an exponential equation as you mentioned previously.

Reviewer #3 (Remarks to the Author):

Overall, the work advances the understanding of global soil organic carbon stock and the created datasets benefit Earth system model benchmarking. However, there are some potential flaws with the methodology that need to be clarified or improved.

- What are the noteworthy results?

The study for the first time quantifies the global distribution of MAOC and POC, covering all except the most arid or glacier-covered biomes. The fitted Random Forest model performs well. The results will be useful for the development and benchmarking of next generation Earth system models.

- Will the work be of significance to the field and related fields? How does it compare to the established literature? If the work is not original, please provide relevant references.

The work is significant compared to the established literature. Past work have created global map of MAOC excluding much of the boreal region and have analyzed the drivers of MAOC and POC in cold regions (e.g. García-Palacios et al. 2024 and Georgiou et al. 2022 cited in the study). The spatial coverages of those works are not as comprehensive as this one.

- Does the work support the conclusions and claims, or is additional evidence needed?

The claim made on lines 52-54 is slightly lacking in support. The cited literature are conceptual models that have not been implemented into equation forms. Are there any implemented models that use MAOC and POC as distinct pools?

It is not clear whether the dots in Fig. 3c and Fig. 3e are fitting results from the lab incubation experiments or from individual pixels in the global maps of MAOC and POC.

- Are there any flaws in the data analysis, interpretation and conclusions? - Do these prohibit publication or require revision?

The Random Forest model uses depth and clay+silt(%) as covariates (Fig. 5). Those results look interesting but are not discussed in the text. It is also surprising that the soil order had no effect - could the author do some more digging to understand why that is the case?

The hyperparameter tuning procedure and the finally list of selected parameters for the Random Forest model are not reported. The method for calculating the predictors' importance is not reported. There are many methods to derive predictors importance (e.g. mean decrease impurity/accuracy, permutation importance, SHapley Additive exPlanations [SHAP]). The SHAP values can also be used to calculate partial dependence plots. The importance ranking and partial dependence might change depending on the methods. Therefore, it is recommended that the authors test the robustness of the results across a few different methods.

The choice of predictors in the Random Forest model requires more justification. Why is fungi:bacteria ratio used in the fitting on turnover rates but not on MAOC and POC? A past study cited inside this paper (Doetterl et al. 2015) used variables like base saturation and CEC - why are those not used in the present study? How about nitrogen deposition (which may stimulate microbial activities) and wetland conditions? How sensitive are the predicted MAOC and POC maps to slight changes in predictor variables?

The abstract gives uncertainty intervals, but the methods section does not document how the uncertainty interval is calculated.

The use of a two-pool model to estimate the turnover rates and carbon use efficiency seems too simplistic. It may be appropriate to fit such a model on the laboratory incubation results, but lines 233-236 further says the model is used to fit the global maps of MAOC and POC. That would certainly require a plant litter input term and perhaps loss via leaching? In fact, the same paragraph says belowground NPP is used, but it cannot be seen how, from the structure of the two-pool model.

The fitting procedure for the two-pool model is not very clearly reported. Suppl. Fig.10 suggests soil respiration data is used, but the corresponding 110-case supplementary data file only shows POC and MAOC values, no soil respiration. Also, one would expect time series data are needed to fit turnover rates, but the corresponding file only shows static values per study. How did the authors fit the two-pool model using static values of the MAOC and POC maps?

- Is the methodology sound? Does the work meet the expected standards in your field?

Yes except for some potential flaws in data analysis noted above.

- Is there enough detail provided in the methods for the work to be reproduced?

Yes.

Dear Reviewers:

Thank you very much for considering the manuscript of ***Global turnover of soil mineral-associated and particulate organic carbon*** (NCOMMS-24-00514). We are grateful for the critical comments and suggestions raised by Reviewers on the manuscript, based on which we have thoroughly revised it. Hope our revision relieved the concerns raised and enhanced the quality of the manuscript. For more details, please refer to point-by-point response to the comments (The original Editors and Reviewers' comments are colored blue). In the revised manuscript, the revised parts have been shown in **red-inked** text.

Reviewer #1 (Remarks to the Author):

The authors conducted an impressive study which they applied a machine learning algorithm with 7163 soil observations combined with a data-model to generate the global MAOC and POC maps, as well as the turnover times of MAOC and POC. The results suggest that turnover times of MAOC and POC are negatively affected by the mean annual temperature, with POC turnover more sensitive to temperature in warm regions but less in cold regions compared to the MAOC turnover. These results are applaudable and will constitute a valuable database for the research community. However, there are some major concerns (particularly in Method section) that need clarification.

Response: Thank you for taking the time to review our manuscript and providing valuable comments, which have been immensely helpful in improving the quality of our manuscript.

Comment 1: L29, "...the vulnerability of soil carbon to warming.", not only to warming, change as "...the vulnerability of soil carbon to climate change.

Response: Thanks, revised.

Comment 2: L52-58, the fates of MAOC and POC are indeed important for Earth system models, but it was not a key point for this paper. I suggest one sentence is enough for expressing the implication of the MAOC and POC to Earth system models

Response: Thanks. We refined as "Incorporating the measurable and biophysically defined POC and MAOC pools into Earth system models probably is an optimal approach to trace the fate of SOC under global change and would be the next generation of soil C cycling model". (L49-52).

Comment 3: L40-71: the two paragraphs are not in good logic; authors did not focus on the knowledge gap in relation with the results later presented. For instance, how climate, plant, soil pH and microbe affect the MAOC and POC pool and turn over? Authors did not mention in this section, but authors talked much more about their

effects on the MAOC and POC dynamics in later parts.

Response: Thanks. We had rewritten the introduction. (L49-72).

Comment 4: L159, please provide a detailed proportion for “most of studies”. Also, I think the authors should provide a scientific mechanism about selecting 53- μm sieve rather than 63- μm sieve.

Response: Thanks for your comments. We have deleted the expression of “most of studies” in the revised manuscript. Based on a recent global synthesis that collected MAOC passed through 63- μm sieve (1144 entries) (Georgiou, K. et al. 2022), we conducted a scientific analysis and found that the sieve size varied greatly across all the entries, where 53- μm sieve size has the largest proportion at 60.1%, followed by 20- μm sieve size (24.1%) and 63- μm sieve size (7.8%) (**showed in the following figure**). To reduce the sieve size impacts from different entries, our datasets just recorded the MAOC passed through 53- μm sieve.

Comment 5: How do you detail with temporal variation of soil sampling? Keep consistent time is important for mapping spatial trends of MAOC and POC. The same issue for the predictors.

Response: Thanks for your critical comments. In the revised manuscript, we had updated our dataset to date, and the data points have been raised from 7163 to 8341 (**Supplementary Fig. 1; Extended Data Datasets 1**). We record the sampling time of each study, which ranged from 1990 to 2022 (**Supplementary Fig. 2**). We also explore the contribution of sampling time on the variances of MAOC and POC by random forest algorithm. We found that sampling time was not important for predicting MAOC and POC (**Supplementary Fig. 3**), meanwhile, sampling time was removed from the best random forest model by “wrapper” methods (**Fig. 1 in revised manuscript**). Soil C is the long-term balance between C inputs from net primary productivity and microbial decomposition, therefore, the averaged NPP across 2001-2023 was used. Considering

the technical challenges for data synthesis, we acknowledged this limitation in the revised manuscript (L152-154).

Supplementary Fig. 2 | The distribution of sampling year.

Supplementary Fig. 3 | The importance of predictors from full models. Two types of importance (impurity and permutation) were first normalized to the interval of [0, 1] according to their maximum and minimum values. The mean of the normalized importance was then calculated. MAT, mean annual temperature; MAP, mean annual precipitation. NPP, net primary productivity. CEC, cation exchange capacity. Bast, base saturation.

Comment 6: L182, there are many machine learning algorithms, why do you select the random forest? How do you identify the optimal hyper-parameters? Please provide the details. How do you evaluate the uncertainties of global maps?

Response: Thanks for your valuable suggestions. First, in the revised manuscript, we compared the performances of six typical machine learning algorithms, including random forest, extreme gradient boosting, support vector machines, recursive partitioning and regression tree, neural networks, and multivariable linear regression. We found that the random forest had the least root mean square error (**Supplementary Table 2**), therefore, random forest was used to general the global maps of MAOC and POC.

Second, the “wrapper” method was used to select the covariates (Wadoux et al. 2020). In specific, by re-calibrating a random forest model several times, each time removing the least important covariate, one may expect to reduce considerably the overall number of covariates with little or no decrease in model prediction accuracy (**L205-218**).

Third, in the revised manuscript, we used the 100-time bootstrapping to generate the uncertainties (**L223-224**).

References

Wadoux, A. M. C., Minasny, B. & McBratney, A. B. Machine learning for digital soil mapping: Applications, challenges and suggested solutions. *Earth-Science Reviews* 210, 103359 (2020).

Supplementary Table 2 | The performance of six machine learning algorithms.
RMSE, root mean square error.

C Fractions	Machine learning algorithms	RMSE
MAOC	Random forest	0.040
	Extreme gradient boosting	0.046
	Support vector machine	0.059
	Recursive partitioning and regression tree	0.077
	Neural network	0.078
	Multivariable linear regression	0.078
POC	Random forest	0.055
	Extreme gradient boosting	0.062
	Support vector machine	0.072
	Recursive partitioning and regression tree	0.102
	Neural network	0.103
	Multivariable linear regression	0.101

Comment 7: L199-229, What are the purpose of this section, and the linkages with previous sections in the methods? Why not use the in-situ data in two-pool model? How do you identify the parameters (e.g., TE) of two-pool model? $R^2 = 0.987$ is very high, how do you avoid the overfitting?

Response: Thank you for the valuable suggestion. We agree that there are gaps between in-situ data and incubation data. We had deleted the incubation data in the revised manuscript. In addition, considering **Reviewer #3's Comment 10**, we had revised the two-pool models by considering the vertical soil C dynamics in field conditions (**Supplementary Fig. 10; L225-241**).

We had generated the MAOC and POC maps for 0–20, 20–40, 40–60, 60–80, and 80–100 cm in the revised manuscript. Referring to Community Land Model (CLM; Koven et al., 2013), an exponential decrease in decomposition was used account for the regulation of soil depth on decomposition:

$$KM_i = KM_1 \exp\left(\frac{z_i - 10}{z_M}\right)$$

$$KP_i = KP_1 \exp\left(\frac{z_i - 10}{z_P}\right)$$

where z_M and z_P are the e-folding depths for C turnovers for MAOC and POC, respectively. z_i is the soil depth of i th layer. KM_1 and KP_1 are MAOC and POC decomposition rates at the top layer, while KM_i and KP_i are MAOC and POC decomposition rates at i th layer. A greater value of e-folding depth for C turnover represents a slower decrease in decomposition rates with increased soil depth, while a smaller value of e-folding depth for C turnover represents a rapider one.

We also found that the e-folding depth for C turnover for POC was about 1.6 times as large as that of MAOC, indicating that increase in soil depth had stronger inhibition on MAOC decomposition than that of POC. MAT predominately control the variations of e-folding depths for C turnovers for both MAOC and POC, implying that climate warming-induced increase in decomposition was greater in subsoil than that in topsoil. (Fig. 5; L132-150).

References

Koven, C. D. et al. The effect of vertically resolved soil biogeochemistry and alternate soil C and N models on C dynamics of CLM4. *Biogeosciences* 10, 7109-7131 (2013).

Supplementary Fig. 10 | Model scheme and validation. a MAOC and POC dynamics model. **b** The relationship between observed C and modelled C.

Fig. 5 Factors influencing the reduce rate of decomposition rate with soil depth. **a** and **b** Importance of predictors for e-folding depths for C turnovers of MAOC and POC, respectively. **c** The e-folding depths for C turnovers of MAOC vs. POC (mean with 5% and 95% quantiles) with paired t -test. **d** Partial dependence showing the effect of mean annual temperature (MAT) on e-folding depths for C turnovers of MAOC and POC. MAP, mean annual precipitation. NPP, net primary productivity. CEC, cation exchange capacity. Bast, base saturation.

Comment 8: Are results consistent for the in-situ data and incubation data? Caution must be exercised for the translation of incubation analyses, which are the conditions used so far for most research, to realistic field conditions.

Response: Thank you for the valuable suggestion. We had deleted the incubation data in the revised manuscript. Please refer to our response to your last comment.

Comment 9: L231-232, I note that the authors did not consider the anthropogenic factors. If consider the anthropogenic factors, here the assumption is wrong. However, the author did not clarify this, for both global maps and two-pool model.

Response: Thanks for your critical comments. When we established the optimal random forest models to predict global patterns of MAOC and POC, the land cover was classified into anthropogenic managed (tundra, boreal forest, temperate conifer forest, temperate broadleaf/mixed forest, (sub)tropical moist forest, (sub)tropical dry forest, montane grassland/shrubland, temperate grassland/shrubland, (sub)tropical grassland/shrubland, Mediterranean, desert, wetland) and natural ecosystem (cropland). The land cover was important for POC but for MAOC (**Figs. 1 and 2**). We also

discussed such effects of land cover and anthropogenic managed in the revised manuscript (L109-112). We also considered the regulation of nitrogen deposition on MAOC and POC (Supplementary Fig. 3).

In the two-pool model, we did assume that the soil C pool is under stable condition but ignored the anthropogenic disturbance. First, we have more robust global estimates of productivity than of heterotrophic respiration. Therefore, if had time series data of both NPP, MAOC, and POC, we did not need the assumption of the steady state. However, despite we provided the big dataset, we did not find significant effect of sampling time on both MAOC and POC at the global scale (Response to Comment 5). Second, the assumption of steady state makes data assimilation computationally more feasible than that under non-steady states (see the non-steady-state data assimilation; Zhou et al., 2013, 2015). Although soil C in ecosystems with anthropogenic disturbance, such as degraded and restored ecosystems, may not be at the steady state, previous studied have shown that such a disequilibrium component is minor for SOC pools considering its long turnover time (Tao et al., 2023; Lu et al., 2018). Finally, we had acknowledged such limitation in the revised manuscript (L152-159), despite the assumption of steady state is the most effective method to answer our questions.

References

- Lu, X., Wang, Y.-P., Luo, Y. & Jiang, L. Ecosystem carbon transit versus turnover times in response to climate warming and rising atmospheric CO₂ concentration. *Biogeosciences* 15, 6559-6572 (2018).
- Tao, F. et al. Microbial carbon use efficiency promotes global soil carbon storage. *Nature* 618, 981-985 (2023).
- Zhou, T., Shi, P., Jia, G. & Luo, Y. Nonsteady state carbon sequestration in forest ecosystems of China estimated by data assimilation. *Journal of Geophysical Research: Biogeosciences* 118, 1369-1384 (2013).
- Zhou, T. et al. Age-dependent forest carbon sink: Estimation via inverse modeling. *Journal of Geophysical Research: Biogeosciences* 120, 2473-2492 (2015).

Comment 10: L255, how do you identify the reference decomposition rate and reference temperature? Please provide the details. How do you perform the translation of incubation analyses to field conditions?

L271, please provide the references to support your assumption (linear model). Because Q₁₀ is traditionally represented by an exponential equation as you mentioned previously.

Response: Thanks for your comments. We had deleted the section of incubation data in the revised manuscript according to your and **Reviewer #3's** suggestions. In addition, thanks for your suggestions and **Reviewer #3's Comment 10**, we had revised the two-pool models by considering the vertical soil C dynamics. We focused on the C turnover of subsoil rather than the climatological Q_{10} in the revised manuscript (please also refer to our **Response to your Comment 7**).

Reviewer #3 (Remarks to the Author):

Comment 1: Overall, the work advances the understanding of global soil organic carbon stock and the created datasets benefit Earth system model benchmarking. However, there are some potential flaws with the methodology that need to be clarified or improved.

Response: Thanks for your support of our study and insight suggestions. We have addressed all the concerns as described below.

Comment 2: - What are the noteworthy results?

The study for the first time quantifies the global distribution of MAOC and POC, covering all except the most arid or glacier-covered biomes. The fitted Random Forest model performs well. The results will be useful for the development and benchmarking of next generation Earth system models.

Response: Thanks for your encouraging comments.

Comment 3: Will the work be of significance to the field and related fields? How does it compare to the established literature? If the work is not original, please provide relevant references. The work is significant compared to the established literature. Past work have created global map of MAOC excluding much of the boreal region and have analyzed the drivers of MAOC and POC in cold regions (e.g. García-Palacios et al. 2024 and Georgiou et al. 2022 cited in the study). The spatial coverages of those works are not as comprehensive as this one.

Response: Thanks for your encouraging comments. In the revised manuscript, we also updated the dataset up to date, and the number of observations were increased from 7163 to 8341.

Comment 4: Does the work support the conclusions and claims, or is additional evidence needed?

The claim made on lines 52-54 is slightly lacking in support. The cited literature are conceptual models that have not been implemented into equation forms. Are there any implemented models that use MAOC and POC as distinct pools?

Response: Thanks for your suggestions, we added another two references:

Abramoff, R. Z. et al. Improved global-scale predictions of soil carbon stocks with Millennial Version 2. *Soil Biology and Biochemistry* 164, 108466 (2022).

Robertson, A. D., Paustian, K., Ogle, S., Wallenstein, M. D., Lugato, E. & Cotrufo, M. F. Unifying soil organic matter formation and persistence frameworks: the MEMS model. *Biogeosciences* 16, 1225-1248 (2019).

Comment 5: It is not clear whether the dots in Fig. 3c and Fig. 3e are fitting results from the lab incubation experiments or from individual pixels in the global maps of MAOC and POC.

Response: Thanks for your suggestions. It is from individual pixels. In addition, we had changed the model (please refer to our **Responses to Reviewer #1's Comment 7 and 10**).

Comment 6: Are there any flaws in the data analysis, interpretation and conclusions?
- Do these prohibit publication or require revision?

The Random Forest model uses depth and clay+silt(%) as covariates (Fig. 5). Those results look interesting but are not discussed in the text. It is also surprising that the soil order had no effect - could the author do some more digging to understand why that is the case?

Response: After updating the dataset and considering the predictors of base saturation, CEC, and nitrogen deposition (proposed from your **Comment 8**), we still found that the soil types was not important for both MAOC and POC (**Fig. 1; Supplementary Fig. 3**). We also improved the discussion of factors influencing MAOC and POC in the revised manuscript (**L83-112**).

Comment 7: The hyperparameter tuning procedure and the finally list of selected parameters for the Random Forest model are not reported. The method for calculating the predictors' importance is not reported. There are many methods to derive predictors importance (e.g. mean decrease impurity/accuracy, permutation importance, SHapley Additive exPlanations [SHAP]). The SHAP values can also be used to calculate partial dependence plots. The importance ranking and partial dependence might change depending on the methods. Therefore, it is recommended that the authors test the robustness of the results across a few different methods.

Response: Thanks for your suggestions. In the revised manuscript, we used methods of impurity and permutation in ranger function of “ranger” package to quantify the importance of predictors, we then normalized the two types of importance to the interval of [0, 1] according to their maximum and minimum values. The mean of the normalized importance was used. We clarified it in the methods (**L210-217**).

Comment 8: The choice of predictors in the Random Forest model requires more justification. Why is fungi:bacteria ratio used in the fitting on turnover rates but not on MAOC and POC? A past study cited inside this paper (Doetterl et al. 2015) used variables like base saturation and CEC - why are those not used in the present study? How about nitrogen deposition (which may stimulate microbial activities) and wetland conditions? How sensitive are the predicted MAOC and POC maps to slight changes in predictor variables?

Response: Thanks for your suggestions. In the revised manuscript, we had considered the predictors of base saturation, CEC, and nitrogen deposition. We explore the effect of wetland conditions by considering the land cover type of wetland (**L188-192**). In addition, we also deleted the microbial variables according to your suggestions (**L183-187; Supplementary Fig. 3**).

Comment 9: The abstract gives uncertainty intervals, but the methods section does not document how the uncertainty interval is calculated.

Response: Thanks for your suggestions. In the revised manuscript, we used the 100-time bootstrapping to generate the uncertainties of global maps. Clarified in the method

(L223-224).

Comment 10: The use of a two-pool model to estimate the turnover rates and carbon use efficiency seems too simplistic. It may be appropriate to fit such a model on the laboratory incubation results, but lines 233-236 further says the model is used to fit the global maps of MAOC and POC. That would certainly require a plant litter input term and perhaps loss via leaching? In fact, the same paragraph says belowground NPP is used, but it cannot be seen how, from the structure of the two-pool model.

Response: Thanks for your suggestions. First, we had deleted the incubation data in the revised manuscript according to your and **Reviewer #1** suggestions.

Second, we had revised the original two-pool model by introducing vertical C dynamics. Referring to Community Land Model (CLM; Koven et al., 2013), the vertical transportations of soil C were represented by diffusion and advection, while the leaching is one process of such vertical transportations. (L225-241).

A recent global study had quantified the allocation of NPP into belowground by soil depth (0–20, 20–40, 40–60, 60–80, 80–100, 100–150 and 150–200 cm; Xiao et al., 2023). Therefore, we firstly obtained the averaged MODIS NPP across 2001 to 2023. We then calculated the NPP allocation into 0–20, 20–40, 40–60, 60–80, and 80–100 cm, respectively. (L229-233).

References

- Koven, C. D. et al. The effect of vertically resolved soil biogeochemistry and alternate soil C and N models on C dynamics of CLM4. *Biogeosciences* 10, 7109-7131 (2013).
- Xiao, L. et al. Global depth distribution of belowground net primary productivity and its drivers. *Global Ecology and Biogeography* 32, 1435-1451 (2023).

Comment 11: The fitting procedure for the two-pool model is not very clearly reported. Suppl. Fig.10 suggests soil respiration data is used, but the corresponding 110-case supplementary data file only shows POC and MAOC values, no soil respiration. Also, one would expect time series data are needed to fit turnover rates, but the corresponding file only shows static values per study. How did the authors fit the two-pool model using static values of the MAOC and POC maps?

Response: Thanks for your critical comments. First, we had deleted the incubation data in the revised manuscript according to your and **Reviewer #1** suggestions.

Second, we have more robust global estimates of productivity than of heterotrophic respiration. Considering the lack of robust time series of heterotrophic respiration, if we assume that soils are approximately at steady state, we can analytically calculate the steady-state pool sizes of MAOC and POC by a semi-analytical solution to accelerate spin-up (Xia et al., 2012; L241-244). More detail please refer to the methods in the revised manuscript.

References

- Xia, J., Luo, Y., Wang, Y.-P., Weng, E. & Hararuk, O. A semi-analytical solution to

accelerate spin-up of a coupled carbon and nitrogen land model to steady state. Geoscientific Model Development 5, 1259-1271 (2012).

Comment 12: Is the methodology sound? Does the work meet the expected standards in your field?

Yes except for some potential flaws in data analysis noted above.

Response: Thanks for your suggestion, more details please refer to our above responses. In addition, we had provided the codes for the data analyses.

Comment 13: Is there enough detail provided in the methods for the work to be reproduced?

Yes.

Response: Thanks for your suggestion.

We hope that you find our revision satisfactory. Thank you very much!

Respectfully,

Chengjie Ren, on behalf of all co-authors

College of Agronomy, Northwest A&F University, Yangling, 712100 Shaanxi, China;

The Research Center of Recycle Agricultural Engineering and Technology of Shaanxi Province, Yangling 712100 Shaanxi, China

Tel: +8613892872667, Fax: +86-87082104; Email: Rencj1991@nwsuaf.edu.cn

REVIEWER COMMENTS

Reviewer #1 (Remarks to the Author):

Authors made detailed responses and corrections based on reviewers' comments and suggestions, and this resubmission is clearly improved. However, I still have two main concerns: (1) authors did not keep consistent time for the input variables (X) and output variables (Y) in the models. (2) It was unclear how authors identified the parameters of two-pool models. In other words, how do authors ensure that the parameters of CLM are suit for your two-pool models?

Reviewer #3 (Remarks to the Author):

Thank you for the considerable efforts in addressing the comments. The manuscript is much clearer and the data analysis more robust. I only have one small suggestion. In the new supplementary information Fig. 5 - 6, please adjust the colorbars to have the same scale within the MAOC group and within the POC group. This will make it easier to see how uncertainty window changes vertically and how the individual layers compare to overall uncertainty.

Dear Reviewers:

Thank you very much for considering the manuscript of *Global turnover of soil mineral-associated and particulate organic carbon (NCOMMS-24-00514)*. We are grateful for the critical comments and suggestions raised by Reviewers on the manuscript, based on which we thoroughly revised it. Hope our revision relieved the concerns raised and enhanced the quality of the manuscript. For more details, please refer to point-by-point response to the comments (The Reviewers' comments are colored blue). In the revised manuscript, the revised parts have been shown in red-inked text.

Reviewer #1 (Remarks to the Author):

Authors made detailed responses and corrections based on reviewers' comments and suggestions, and this resubmission is clearly improved. However, I still have two main concerns:

(1) authors did not keep consistent time for the input variables (X) and output variables (Y) in the models.

Response: Thank you for taking the time to review our manuscript and providing valuable comments again.

Most of the data entries were from 2000-2021 (Supplementary Fig. 2), together with the time range of MODIS net primary productivity, we calculated the average annual temperature, precipitation, and net primary productivity across 2001 to 2021. Therefore, we had kept the consistent times for climates, net primary productivity, MAOC, and POC. Correspondingly, we had conducted all following analyses (L191-195), the original conclusion was not changed.

The climates variables in last version were from climate data for 1970-2000. MAT had negative effect on both MAOC and POC, therefore, it is expected that our original estimations of MAOC and POC storages and turnover times were higher than the current estimations. Your critical comments had increased the accuracy of our global estimations.

Supplementary Fig. 2 | The distribution of sampling year.

(2) It was unclear how authors identified the parameters of two-pool models. In other words, how do authors ensure that the parameters of CLM are suitable for your two-pool models?

Response: Sorry for our confused expression in the last version of manuscript. Here, we did not use the constant parameters from previous models. Instead, the data assimilation approach was used to obtain the optimal model parameters based on the prior range of the model parameters (we added the **Supplementary Table 3** showing the ranges). We also clarified it in the method section (**L261-274; L284-285; L290-291**).

Reviewer #3 (Remarks to the Author):

Thank you for the considerable efforts in addressing the comments. The manuscript is much clearer and the data analysis more robust. I only have one small suggestion. In the new supplementary information Fig. 5 - 6, please adjust the colorbars to have the same scale within the MAOC group and within the POC group. This will make it easier to see how uncertainty window changes vertically and how the individual layers compare to overall uncertainty.

Response: Thanks for your support of our study. We have revised the figures.

We hope that you find our revision satisfactory. Thank you very much!

Respectfully,

Chengjie Ren, on behalf of all co-authors

State key Laboratory for Crop Stress Resistance and High-Efficiency Production, College of Agronomy, Northwest A&F University, Yangling, Shaanxi, China

Tel: +8613892872667, Fax: +86-87082104; Email: Rencj1991@nwsuaf.edu.cn

REVIEWERS' COMMENTS

Reviewer #1 (Remarks to the Author):

No further comments, except I note that authors had kept the consistent times for climates, net primary productivity, MAOC, and POC. However, soil properties (soil types, pH, clay plus silt content, base saturation, and cation exchange capacity) and other variables were not.